# Predicting the Pathway Involvement of Metabolites Based on Combined Metabolite and Pathway Features

**DOI:** 10.3390/metabo14050266

**Published:** 2024-05-07

**Authors:** Erik D. Huckvale, Hunter N. B. Moseley

**Affiliations:** 1Markey Cancer Center, University of Kentucky, Lexington, KY 40506, USA; edhu227@uky.edu; 2Superfund Research Center, University of Kentucky, Lexington, KY 40506, USA; 3Department of Toxicology and Cancer Biology, University of Kentucky, Lexington, KY 40536, USA; 4Department of Molecular and Cellular Biochemistry, University of Kentucky, Lexington, KY 40506, USA; 5Institute for Biomedical Informatics, University of Kentucky, Lexington, KY 40506, USA

**Keywords:** metabolism, metabolite, metabolic pathway, machine learning, deep learning, XGBoost, multilayer perceptron, supervised learning, binary classification, kyoto encyclopedia of gene and genomes (KEGG)

## Abstract

A major limitation of most metabolomics datasets is the sparsity of pathway annotations for detected metabolites. It is common for less than half of the identified metabolites in these datasets to have a known metabolic pathway involvement. Trying to address this limitation, machine learning models have been developed to predict the association of a metabolite with a “pathway category”, as defined by a metabolic knowledge base like KEGG. Past models were implemented as a single binary classifier specific to a single pathway category, requiring a set of binary classifiers for generating the predictions for multiple pathway categories. This past approach multiplied the computational resources necessary for training while diluting the positive entries in the gold standard datasets needed for training. To address these limitations, we propose a generalization of the metabolic pathway prediction problem using a single binary classifier that accepts the features both representing a metabolite and representing a pathway category and then predicts whether the given metabolite is involved in the corresponding pathway category. We demonstrate that this metabolite–pathway features pair approach not only outperforms the combined performance of training separate binary classifiers but demonstrates an order of magnitude improvement in robustness: a Matthews correlation coefficient of 0.784 ± 0.013 versus 0.768 ± 0.154.

## 1. Introduction

Metabolism is the set of biochemical processes within cells and organisms that sustain life. Metabolites are chemical compounds that take part as the reactants and/or products of chemical reactions involved in metabolism. The products of one reaction can act as reactants in another, resulting in chains of reactions occurring in different parts of a cell or organism, diverging in different directions and serving different metabolic purposes. These reaction chains are organized into networks of connected reactions known as biochemical pathways. These pathways are grouped into broader pathway categories, classified by the types of reactants and products involved, cellular location, metabolic purpose, etc. Since metabolites are the reactants and products of metabolic reactions and reactions are the building blocks of pathways, certain metabolites are necessarily involved in certain pathway categories. However, the entirety of metabolic pathways are not known, because not all of the chemical reactions performed in cellular metabolism have been discovered. This results in a recurring problem faced by biologists, biochemists, and bioinformaticians, which is encountering metabolites and lacking information related to their pathway involvement.

Knowledge bases such as the Kyoto Encyclopedia of Genes and Genomes (KEGG) [1,2,3] and BioCyc, particularly via their database called MetaCyc [4], have made tremendous contributions to creating databases containing metabolites with pathway annotations. However, due to the costly and laborious nature of experimentally determining the pathway involvement of metabolites, many of the metabolites in such metabolic knowledge bases are still not annotated or are only partially annotated to metabolic pathways. For example, as of 3 July 2023, a total of 19,119 compounds existed in the KEGG database with only 6736 of them having an annotated pathway involvement [5]. Because of the lack of annotation, several machine learning methods have been proposed to predict the pathway categories that a metabolite is involved in given information about the metabolite’s chemical structure. There have been several publications in particular that have trained supervised learning models, most notably graph neural networks [6], on compounds in the KEGG database. The models were trained and evaluated on datasets with metabolites as entries, with the features being the chemical structure information of these metabolites and the labels being the high-level metabolic pathway categories. Specifically, KEGG provides a hierarchy of broader pathway categories branching into more granular pathway categories as seen here: https://www.genome.jp/brite/br08901 (accessed on 3 April 2024). There are 12 categories under metabolism, namely, 1. amino acid metabolism; 2. biosynthesis of other secondary metabolites; 3. carbohydrate metabolism; 4. chemical structure transformation maps; 5. energy metabolism; 6. glycan biosynthesis and metabolism; 7. lipid metabolism, 8. metabolism of cofactors and vitamins; 9. metabolism of other amino acids; 10. metabolism of terpenoids and polyketides; 11. nucleotide metabolism; 12. xenobiotics biodegradation and metabolism. Several past publications trained models to predict 11 out of the 12 categories, excluding ‘Chemical structure transformation maps’, likely due to its difficulty to predict. Since some of these publications were proven to be invalid, using a dataset that contained duplicate entries in both the train and test sets [7], Huckvale et al. developed a new KEGG-derived benchmark dataset for the task of developing models for predicting the pathway category involvement for all 12 of the aforementioned categories based on the metabolite chemical structure [5]. Huckvale et al. also demonstrated a set of binary classification models trained on this benchmark dataset for predicting pathway category involvement.

However, these benchmark models involved training a separate model for each pathway category. This approach complicates the design, implementation, and maintenance of the model training, evaluation, and deployment pipeline. The amount of computational resources required are multiplied by the number of models needed, which is one for each pathway category. These shortcomings are exacerbated further when it is proposed that the models are trained on more granularly defined pathway categories, compared to the most high-level pathway categories. For example, KEGG defines 12 topmost pathway categories but defines 184 in total when descending one level down the hierarchy. The current benchmark dataset includes 5683 metabolites with pathway annotations and chemical structure representations, which is adequate to train 12 pathway category-specific models but is woefully inadequate to train 184 separate models, due to the diminishingly small number of positive entries for each pathway. Additionally, the resulting models are only equipped to predict the specific pathway category they were trained on. There are several other databases and use cases that may not organize the pathway category in the exact same way as the hierarchy found in KEGG. For example, we see that MetaCyc provides an entirely different pathway hierarchy as seen here: https://metacyc.org/META/class-tree?object=Pathways (accessed on 3 April 2024). This suggests the need for a more generic model.

In this work, we present a single binary classifier for predicting the pathway involvement of metabolites. The dataset provided by the work of Huckvale et al. [5] uses an atom coloring technique [8] to represent the substructures of molecules, with the features of a metabolite being the count of such molecular substructures that are present in the molecule. Every pathway category has certain metabolites associated with it, so a pathway category can be generically represented by aggregations of the features of the metabolites associated with the pathway category. Given the chemical structure information of the metabolites, along with structured informational representations of the pathway categories, we have constructed a dataset where the entries consist of metabolite features paired with pathway features. With a dataset of these metabolite–pathway features pairs, we have trained and evaluated the models to predict whether the metabolite of the pair is involved in the pathway category of the pair. This approach uses just one model that can effectively predict the original 12 pathway categories and paves the way for a generic classifier that can predict the presence of metabolites in arbitrary pathway categories. We demonstrate that not only does the metabolite–pathway pairs approach perform well compared to training a separate model per classifier, but our best model (a multilayer perceptron) has even exceeded the combined performance of the prior benchmark models (Random Forest, XGBoost, and multilayer perceptron models), not just in the average Matthews correlation coefficient (MCC) but also with a significantly lower standard deviation.

## 2. Materials and Methods

### 2.1. Generating the Feature Vectors

Huckvale et al. [5] previously generated a dataset of 5683 entries, with each entry containing a vector of the atom color [8] features corresponding to a metabolite. Building off of this dataset of metabolite features, we constructed pathway features via the process in Figure 1. Each of the 12 pathway categories have a subset of the 5683 available metabolites that are associated with it (e.g., ‘Amino acid metabolism’ contains 611 out of the 5683 metabolites, ‘Biosynthesis of other secondary metabolites’ contains a different subset of 1486 metabolites, etc.). With each subset, we created vectors that are sums of the features of the metabolites associated with the corresponding pathway category. While the features of a single metabolite are the counts of the atom colors within it, the resulting summed features are the number of occurrences of the atom colors across all the metabolites within the entire pathway category. However, we could not simply use these raw counts for the pathway features, because different pathway categories have different amounts of metabolites within them. To correct this, we applied a bond inclusivity-specific soft max normalization by calculating the proportion of each atom color within each pathway category and dividing the raw counts by the total count of the other atom colors within the same pathway category and of the same level of bond inclusion (Figure 1). For example, for the elemental atom colors (0-bond-inclusion), if the total count of elemental atom colors for a given pathway category was 10,000 but the count for the carbon atom was 1000, then the feature value for the carbon atom color in that pathway category would be 0.1. This effectively normalizes the pathway categories, such that they become comparable to each other despite one category having more metabolites than another. For consistency, we also normalized the metabolite features in the same way. The resulting metabolite features were the proportion of occurrences of each atom color compared to every other atom color of the same bond inclusion level within a single compound and the resulting pathway features were likewise the proportion of occurrences of that atom color across the compounds within the entire pathway category (Figure 1).

Finally, upon creating the 12 pathway feature vectors, there were duplicate pathway features across the 12 vectors. We created a copy of the set of pathway features with the duplicate features removed. The original set (containing the duplicate features) was used to make the encoded features using an autoencoder (Figure 1) while the de-duplicated set was used to train the models directly. While autoencoders [9] have been known to improve the classification performance via feature reduction, particularly reducing redundant features, they generally reduce training time and other computational resources. We created encoded counterparts of the metabolite and pathway feature sets to determine whether our models would perform at least as well after passing through an autoencoder. While we initially normalized entry-wise, we additionally normalized feature-wise prior to training the autoencoder via min–max scaling (Figure 1). The autoencoder was trained on both the metabolite and pathway features since they had the same atom colors (the original set of pathway features was used by the autoencoder while the de-duplicated set of pathway features was used downstream). We performed the min–max scaling again after passing the non-encoded data through the autoencoder to create the final encoded features (Figure 1).

Table 1 provides the characteristics of the individual metabolite feature sets and pathway feature sets before they were paired together. Since we developed the work of Huckvale et al. [5], the number of metabolite entries and features were the same as with their work, with the metabolite features having already been de-duplicated. Since the pathway features were derived from the metabolite features, the number of pathway features was initially the same as that of the metabolite features. De-duplicating the pathway features removed 9220 duplicate features from the original 14,655. Such a large proportion of the features were duplicates likely because we were considering just 12 pathway categories compared to 5683 metabolites. Since the pathway features were not de-duplicated before passing through the autoencoder, the autoencoder could encode both the pathway features and metabolite features (autoencoders expect their input to always be the same size). Encoding the features to one-tenth their original size resulted in both sets having 1465 encoded features (Table 1).

### 2.2. Dataset Engineering of the Metabolite–Pathway Features Pair Dataset via Cross Join

Once all four feature sets were complete, we fed the data to the machine learning models as a cross join between the metabolite feature vectors and the pathway feature vectors (i.e., every metabolite feature vector was joined with every pathway feature vector). Concatenating the two vectors together resulted in sets of metabolite–pathway features pair entries (Figure 2). The corresponding label of each entry was a binary label indicating whether the given metabolite was part of the given pathway category (a positive entry) or not (a negative entry).

Table 2 shows that the characteristics of the non-encoded metabolite–pathway features pair dataset were all the same as the encoded counterpart except for the number of features, which, of course, were one-tenth the amount of the non-encoded. The resulting number of entries was the number of metabolite entries (5683) multiplied by the number of pathway entries (12). The number of positive entries was the sum of the number of metabolites associated with each of the 12 pathway categories. While each pathway category had a different number of metabolites associated, the combination of them all resulted in an overall proportion of about 10.6% positive entries (Table 2).

### 2.3. Hyperparameter Tuning and Model Evaluation

While the XGBoost [10] model previously performed best overall in the work of Huckvale et al. [5], we suspected that a neural network approach may perform better, considering the increase in the data as a result of the cross join (68,196 entries compared to 5683). Therefore, we ran experiments using both an XGBoost model as well as a multilayer perceptron (MLP) [11]. Figure 3 shows an overview of the hyperparameter tuning, model training, and model evaluation using both the non-encoded and encoded features. This included performing 100 trials of hyperparameter tuning for each of the four combinations of model and feature sets, using the Optuna Python library [12]. For each trial, we performed up to 20 cross-validation (CV) iterations (some trials had less than 20 if they were pruned due to not showing promise), creating a stratified train–test split [13] for each iteration. With a fairly low proportion of positive entries (Table 2), we decided to duplicate the positive entries in the training sets until the proportion of positive entries was equal or just under 50%. The test sets, however, retained the same proportion of positive entries, since duplicate entries in a test set can lead to overly optimistic and otherwise misleading results. Upon training the model, we obtained predictions on the test set and compared them against the labels of the same, calculating the Matthews correlation coefficient (MCC) [14,15] of each train–test split. The median MCC value across the CV iterations was used to determine the most successful hyperparameter tuning trial. Appendix A shows the hyperparameters selected from the best trial for each model–feature set combination.

Using the best hyperparameters for each model, we trained the models over 1000 CV iterations, performing train–test splits similar to those made during the hyperparameter tuning but performing 1000 iterations instead of 20 for the final evaluation. Pragmatically, we only used 20 iterations for the hyperparameter tuning to save time, expecting that 20 would provide a reasonable estimate of the overall model performance. We calculated five model performance metrics on the test set for each CV iteration including the accuracy, precision, recall, F1 score, and MCC. Additionally, we measured the importance of each input feature, but only for the XGBoost when trained on the non-encoded feature set. We stored the model performance scores and the feature importance scores in a database file to summarize and visualize the results downstream (Figure 3).

For tuning the hyperparameters and the final CV analyses of both the XGBoost and MLP, we used high-performance computing (HPC) machines with a system capacity of 187 GB of RAM and 32 cores per node, with the CPUs being ‘Intel^®^ Xeon^®^ Gold 6130 CPU@2.10GHz (Santa Clara, CA USA)’. Using the SLURM HPC job manager, no more than 72 h of compute time was allocated for each of the four hyperparameter tunings and CV analyses. The XGBoost runs allocated 10 cores with 17 gigabytes of RAM allocated per core. The MLP runs allocated 16 cores with 6 gigabytes of RAM per core. Both of the XGBoost runs used a GPU with 12 GB of GPU memory, with the name of the GPU card being ‘Tesla P100 PCIe 12 GB (Santa Clara, CA USA)’.

All scripts for the data processing and analysis were written in the Python programming language [16], and the results were stored in an SQL [17] database managed by the DuckDB Python package [18]. The summarization and visualization of these results were performed using the Tableau business intelligence software [19] and the Seaborn [20] Python package (built on the MatPlotLib [21] python package) within Jupyter notebooks [22]. The data processing was facilitated by the NumPy [23], Pandas [24], and H5Py [25] Python packages. The evaluation metrics were computed using the Sci-Kit Learn [26] Python package. Pearson and Spearman correlation coefficients were computed using the SciPy [27] Python package. The XGBoost model was implemented using the XGBoost Python package [10], while the autoencoder and MLP were implemented using the Pytorch Lightning [28] and Torch Geometric [29] packages built on top of the PyTorch [30] package. The computational resource profiling was performed using the gpu-tracker Python package [31].

## 3. Results

### 3.1. Model Performance

Table 3 provides the average and standard deviation of the MCC for each combination of model (MLP and XGBoost) and feature set (encoded by the autoencoder or not) and for each of the 12 pathway categories. See Appendix A for all metrics in addition to the MCC.

Figure 4 provides a violin plot of the MCC obtained over the 1000 CV iterations for the MLP models for each pathway category. The distribution of the performance of the model trained on the non-encoded feature set is shown side-by-side with that of the corresponding model trained on the encoded feature set. We see that for most pathway categories, the model trained on the non-encoded set outperformed the encoded counterpart. However, for a few pathway categories, namely, ‘Chemical structure transformation maps’, ‘Glycan biosynthesis and metabolism’, and ‘Energy metabolism’, the MLP trained on the encoded set performed better.

Figure 5 provides the same as Figure 4 but for the XGBoost models. We see that the XGBoost consistently performs significantly worse when trained on the encoded feature set.

Table 4 compares the average and standard deviation MCC of each model and feature set combination. When compared to the previous work of Huckvale et al. [5], which trained a separate binary classifier for each pathway category, we see that the MLP greatly improved when training a single binary classifier on the metabolite–pathway features pairs. The MLP trained on the encoded data performed well compared to the XGBoost model trained on the non-encoded data, which likewise performed well compared to the XGBoost of the previous work. The XGBoost trained on the encoded data performed significantly worse than all other models and is included for completeness. While only the MLP trained on the non-encoded feature set significantly exceeded the average MCC of the best model in the previous work, the standard deviation of all metabolite–pathway pair models were lower by an order of magnitude from those of the previous work.

Figure 6 provides violin plots showing the distribution of the models across all pathway categories. We see that the distributions for the XGBoost trained on the non-encoded data do not even overlap with those trained on the encoded data. For the MLP, the two distributions do overlap, but the MLP trained on the non-encoded data is clearly the highest performing.

To measure the computational resource usage of each model and for each feature set, we performed a subset of the CV iterations, this time while profiling the maximum RAM usage, the maximum GPU RAM, and the real compute time over 50 CV iterations. Table 5 details this information. For example, we see it took about 129 min for the MLP to complete the 50 iterations when training on the encoded data, while taking about 90 min to do the same on the non-encoded data. The XGBoost took less time than the MLP when training on the same data. The non-encoded data moderately increased the GPU RAM and RAM utilization compared to the encoded. The XGBoost required significantly more RAM and an order of magnitude more GPU RAM.

### 3.2. Feature Importance

Using the XGBoost model trained on the non-encoded feature set, we were able to compute the importance of each feature for each CV iteration. The softmax of the feature scores provided each feature importance relative to every other feature, rather than the raw feature importance values. Taking the average relative feature importance across all CV iterations, every feature obtained an overall score indicating its feature importance. Those features scoring 0 were excluded from this analysis (Figure 7).

Table 6 shows the atom colors [8] of the top 10 most important features, specifying whether the atom color corresponds to the pathway features or to the metabolite features. We see that most of the top 10 were pathway features. Associated pathways are defined by the atom color corresponding to a metabolite that exists in the pathway category. For example, the most important feature (rank 1) is associated with the ‘Biosynthesis of other secondary metabolites’, ‘Metabolism of terpenoids and polyketides’, and ’Xenobiotics biodegradation and metabolism’ pathway categories. The rank of the feature is provided with 1 being the single most important feature down to the 10th most important feature. If it is a pathway feature, its pathway rank is the same as its rank. However, the corresponding metabolite feature (the same atom color but representing metabolites instead of pathway categories) has its own rank. Both the pathway and metabolite ranks are not always available, either because the pathway feature corresponding to a metabolite feature may have been excluded from the feature set, having duplicate values as another feature, or because one of the features may have been a part of the feature set, not being a duplicate but consistently scoring 0, and was not considered for feature importance. We see that for every pathway feature, the rank of its corresponding metabolite feature differs significantly.

Figure 8 shows examples of compounds that contain the atom color corresponding to the top 10 most important features. The red highlighted portion contains the atoms and bonds corresponding to the atom color. We see the most important feature is a hydroxyl group connected to a methyl group by a ring structure.

## 4. Discussion

Past publications interested in predicting the pathway involvement of metabolites have trained separate binary classifiers for each specific pathway category in question, which has a number of shortcomings. In this work, we overcame those shortcomings by training a single binary classifier that can predict the presence of a metabolite in a generically described pathway category. The metabolite–pathway pair approach utilizes a combination of feature and dataset engineering methods to increase the resulting dataset over 10-fold from the original gold standard dataset, improving the model training and evaluation, simplifying the deployment process to a single binary classifier, and enabling the possibility of predicting involvement with arbitrarily defined pathway categories. From a search of the literature, we found a prior publication that utilized this metabolite–pathway pair approach for designing a generic binary classifier for a different but similar classification problem predicting the metabolite pathway involvement based on protein–protein interaction network data [32]. In our application predicting the metabolite pathway involvement based on the chemical structure, not only does the resulting single classifier perform well compared to the combined performance of the separate classifiers, our best model even exceeded the performance of the prior benchmark models (Table 4). In particular, our best model demonstrates a greatly improved performance robustness, as evidenced by the significant reduction in the standard deviation by an order of magnitude: a mean MCC of 0.784 ± 0.013 versus 0.768 ± 0.154. We can confidently conclude that the use of the metabolite–pathway features pair entries is a superior method for predicting the pathway involvement based on chemical structure-derived features.

While the XGBoost unsurprisingly performs relatively poorly when trained on the autoencoded features, we see that the XGBoost trained on the non-encoded features performs well compared with the MLP trained on the encoded features. The MLP, when trained on the non-encoded features, exceeds the performance of both, as well as the combined performance of the prior benchmark models (Table 4). While the MLP requires a moderately larger amount of time to train and evaluate than the XGBoost, it requires only a fraction of the RAM and GPU RAM. Furthermore, while the MLP trained on the non-encoded features requires a moderately larger amount of RAM and GPU RAM than that trained on the encoded features, the encoded variant actually requires significantly more time (Table 5). This is likely because, while the non-encoded MLP takes longer to complete an individual epoch, the encoded MLP requires many more epochs to converge. Encoding the data saved RAM, but even the non-encoded variant required less than 3 gigabytes of RAM and less than 1 gigabyte of GPU RAM, an extremely minor amount for modern HPC systems. With time being the more precious resource, we can confidently conclude that training the MLP on the non-encoded data is the best pipeline for this machine learning task, and it appears that the autoencoder did not meaningfully save computational resources, as initially expected.

When considering the importance of the atom color features, there does not appear to be any correlation between a pathway feature and the corresponding metabolite feature of the same atom color, and vice versa (Appendix A). Most of the top 10 features were pathway features, suggesting that pathway information is often more important when predicting the pathway involvement of a metabolite, compared to information about the metabolite itself, at least from the perspective of the XGBoost training.

Since the metabolite–pathway pair approach is designed to scale to an arbitrarily high number of pathway categories, it paves the way for predicting more granular pathway categories, compared to the topmost 12 categories in the KEGG hierarchy. Beyond that, a model can be trained on the pathway categories from KEGG combined with those from MetaCyc, any data source, or any collection of arbitrary pathway categories, since the model is not tied to any particular category or restricted set of categories. This opens up the possibility of combining the datasets derived from both KEGG and MetaCyc for future model development. Moreover, due to the multiplicative effect of the cross join when constructing the metabolite–pathway pairs, the dataset multiplies in size when adding more pathway categories. While this is typically a benefit for improving the performance of the model, it may become impractical to train the XGBoost models, because they require that the entire dataset fits into the memory. However, neural networks train in mini-batches, making the MLP perhaps the more feasible model to use on larger datasets in the future, especially considering that the MLP requires less RAM and GPU RAM and outperforms the XGBoost when given sufficient data. For smaller datasets, such as the one in this work, the XGBoost model may be preferred if analyzing the importance of the features is desired.

## 5. Conclusions

This work uses both feature and dataset engineering to create an over 10-fold larger dataset of metabolite-pathway feature vectors and to train a single binary classifier for predicting KEGG metabolic pathway involvement from chemical structure-derived features. This metabolite–pathway features pair approach outperforms prior machine learning approaches, demonstrating an order of magnitude improvement in robustness: MCC of 0.784 ± 0.013 versus 0.768 ± 0.154. Moreover, this approach paves the way for developing models that can predict involvement with metabolic pathways defined at different levels of metabolic granularity.

## Figures and Tables

**Figure 1 metabolites-14-00266-f001:**
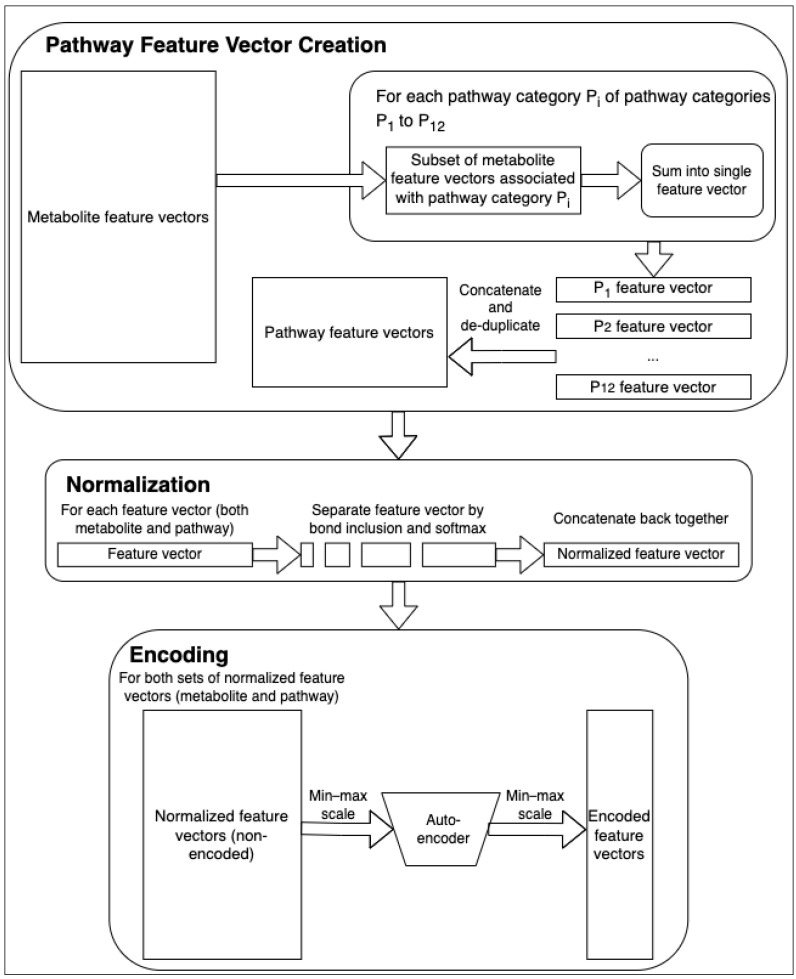
Feature engineering. The starting metabolite atom coloring feature vectors are summed and normalized into the pathway feature vectors. Then, both the metabolite and pathway feature vectors are used to train an autoencoder and to generate encoded feature vectors with a reduced number of embedded features.

**Figure 2 metabolites-14-00266-f002:**
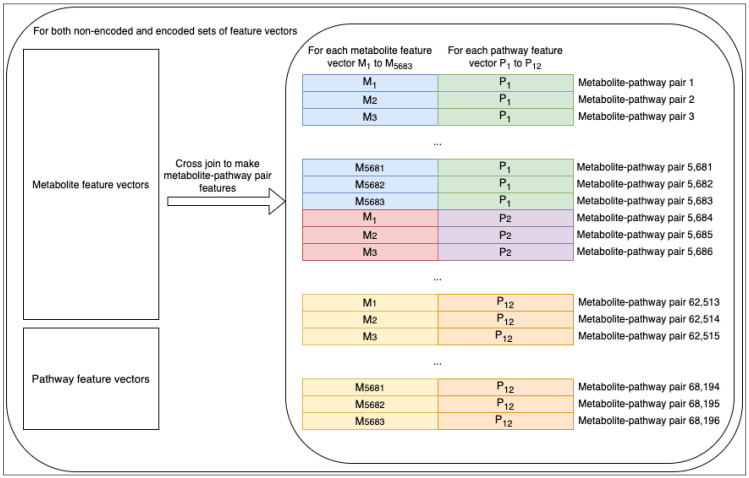
Dataset engineering. In total, 5683 metabolite and 12 pathway feature vectors were cross-joined to create 68,196 metabolite–pathway features pair vectors.

**Figure 3 metabolites-14-00266-f003:**
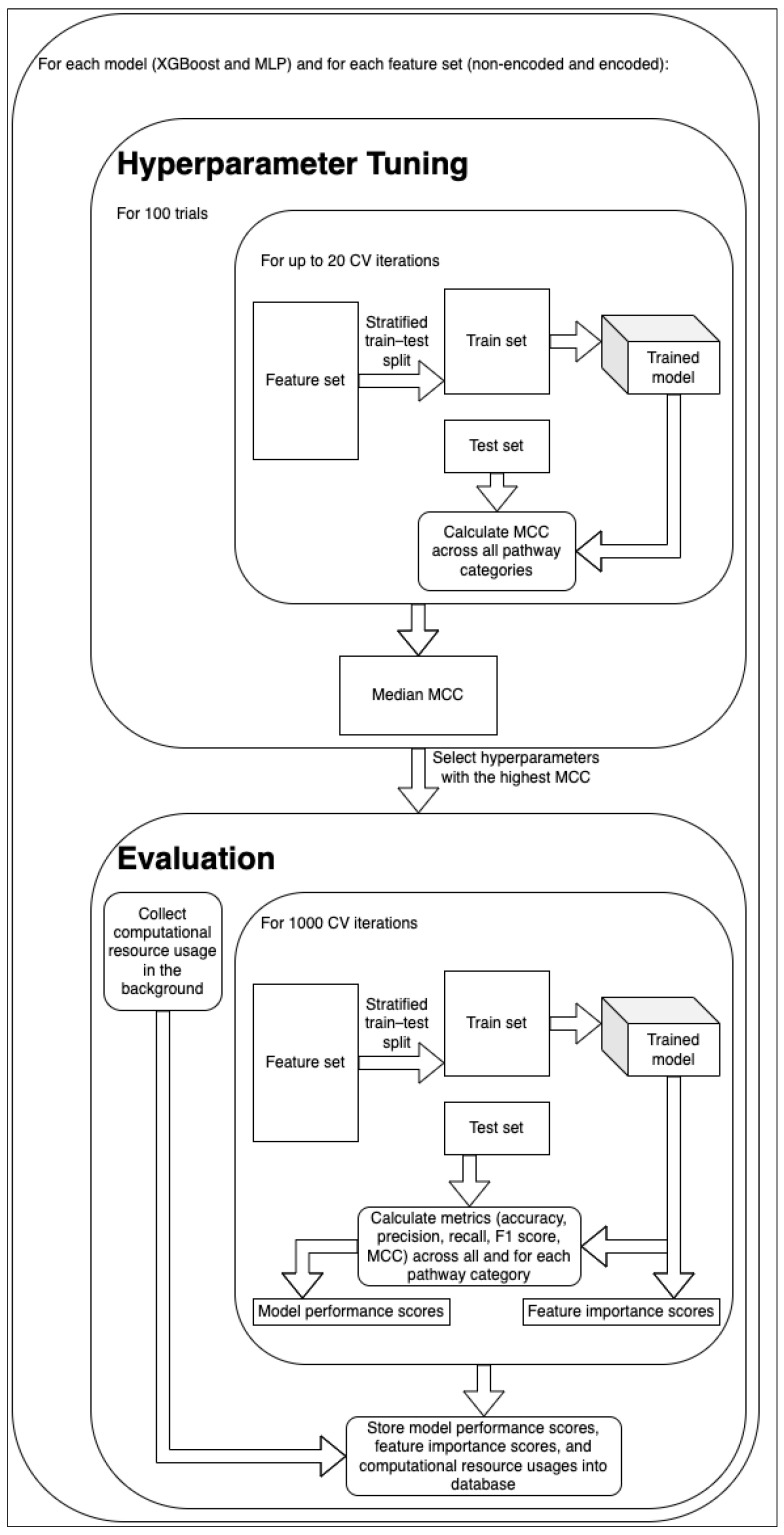
Overview of hyperparameter tuning, model training, and model evaluation. The optimal hyperparameters were selected using 100 trials of hyperparameter tuning and then used in comprehensive model training and evaluation using 1000 CV iterations.

**Figure 4 metabolites-14-00266-f004:**
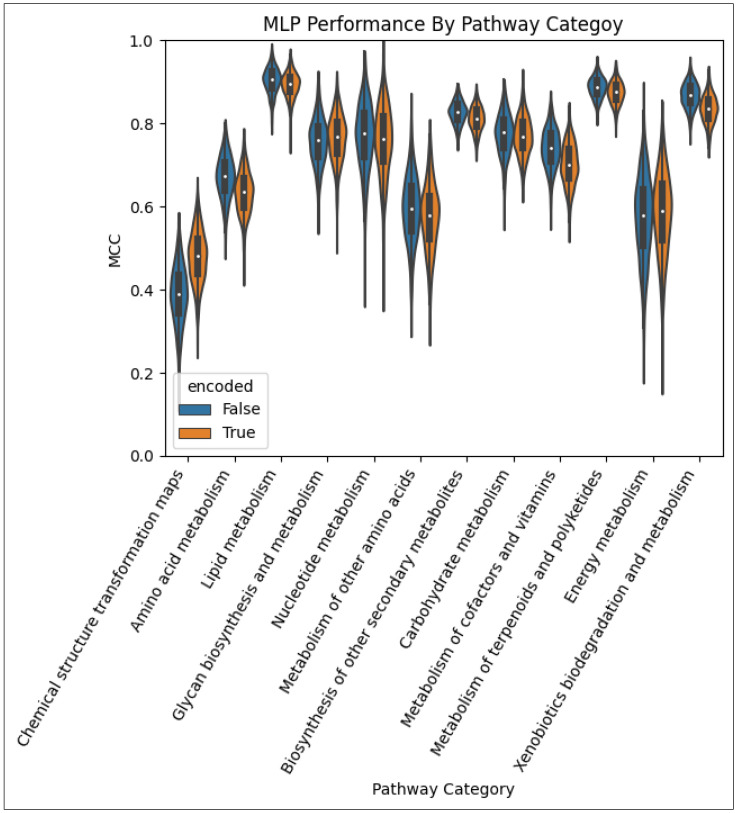
Performance of the MLP models by pathway category. The models using the autoencoded metabolite–pathway features pair vectors are in orange and the models using the unencoded metabolite–pathway features pair vectors are in blue.

**Figure 5 metabolites-14-00266-f005:**
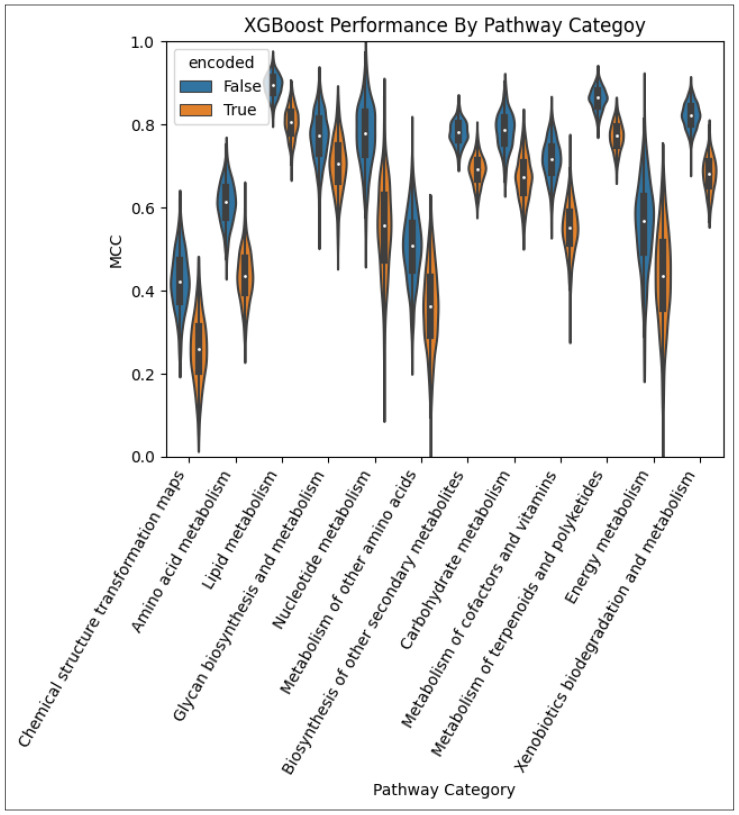
Performance of the XGBoost models by pathway category. The models using the autoencoded metabolite–pathway features pair vectors are in orange and the models using the unencoded metabolite–pathway features pair vectors are in blue.

**Figure 6 metabolites-14-00266-f006:**
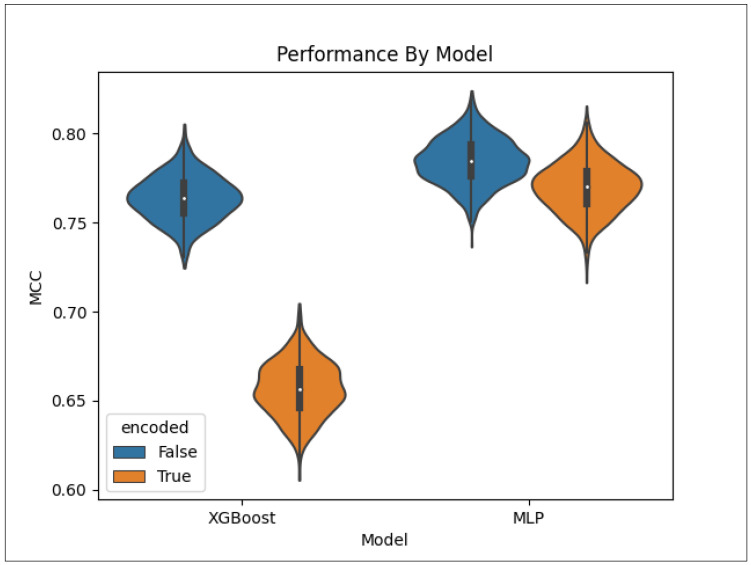
Overall performance by model. The models using the autoencoded metabolite–pathway features pair vectors are in orange and the models using the unencoded metabolite–pathway features pair vectors are in blue.

**Figure 7 metabolites-14-00266-f007:**
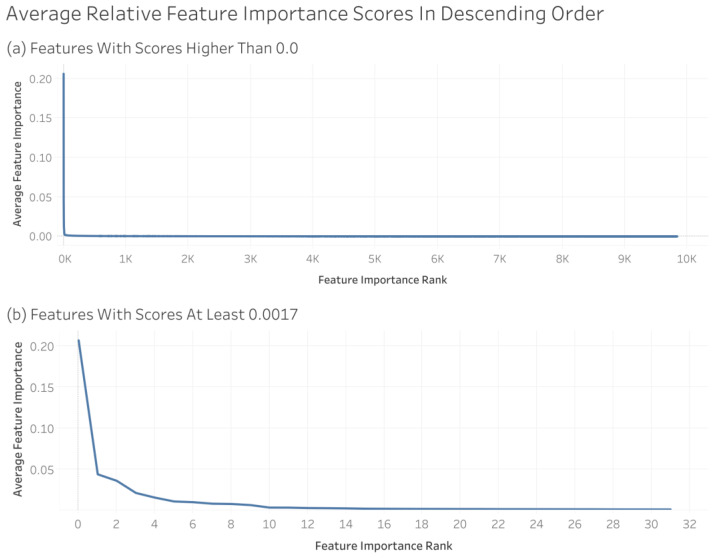
Average relative feature importance scores in descending order. Plot (**a**) includes all features, while plot (**b**) only includes features with a relative feature importance of 0.0017 or greater.

**Figure 8 metabolites-14-00266-f008:**
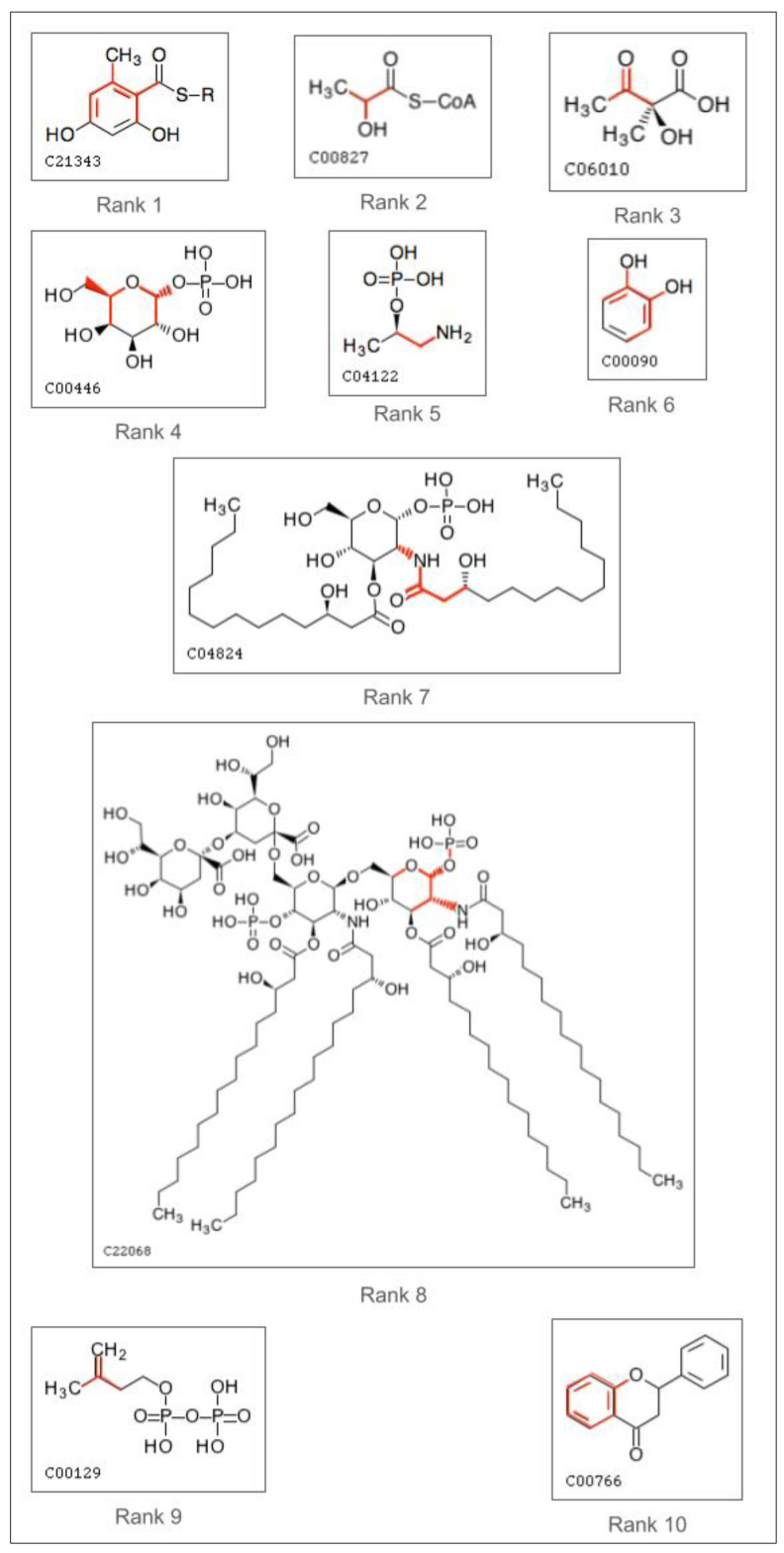
Molecular structures of top 10 atom coloring features. The atoms and bonds comprising the atom coloring feature are highlighted in red.

**Table 1 metabolites-14-00266-t001:** Characteristics of the individual feature sets.

Feature Type	Encoded	Number of Entries	Number of Features
Metabolite	No	5683	14,655
Metabolite	Yes	5683	1465
Pathway	No	12	5435
Pathway	Yes	12	1465

**Table 2 metabolites-14-00266-t002:** Characteristics of the metabolite–pathway features pair datasets after the cross join.

Encoded	Number of Entries	Number of Features	Number of Positive Entries
No	68,196	20,090	7246
Yes	68,196	2930	7246

**Table 3 metabolites-14-00266-t003:** Model performance by pathway category.

Model	Pathway Category	Average MCC	Standard Deviation
MLP	Amino acid metabolism	0.6712	0.0518
	Biosynthesis of other secondary metabolites	0.8268	0.0268
	Carbohydrate metabolism	0.7736	0.0504
	Chemical structure transformation maps	0.3879	0.0696
	Energy metabolism	0.5707	0.1035
	Glycan biosynthesis and metabolism	0.7560	0.0589
	Lipid metabolism	0.9029	0.0320
	Metabolism of cofactors and vitamins	0.7403	0.0481
	Metabolism of other amino acids	0.5937	0.0813
	Metabolism of terpenoids and polyketides	0.8867	0.0251
	Nucleotide metabolism	0.7680	0.0816
	Xenobiotics biodegradation and metabolism	0.8677	0.0305
MLP	Amino acid metabolism	0.6327	0.0520
with	Biosynthesis of other secondary metabolites	0.8110	0.0296
Encoding	Carbohydrate metabolism	0.7694	0.0467
	Chemical structure transformation maps	0.4789	0.0621
	Energy metabolism	0.5814	0.0982
	Glycan biosynthesis and metabolism	0.7650	0.0565
	Lipid metabolism	0.8930	0.0292
	Metabolism of cofactors and vitamins	0.7001	0.0517
	Metabolism of other amino acids	0.5733	0.0812
	Metabolism of terpenoids and polyketides	0.8741	0.0275
	Nucleotide metabolism	0.7571	0.0856
	Xenobiotics biodegradation and metabolism	0.8339	0.0347
XGBoost	Amino acid metabolism	0.6110	0.0525
	Biosynthesis of other secondary metabolites	0.7803	0.0293
	Carbohydrate metabolism	0.7857	0.0454
	Chemical structure transformation maps	0.4221	0.0722
	Energy metabolism	0.5578	0.1090
	Glycan biosynthesis and metabolism	0.7701	0.0599
	Lipid metabolism	0.8938	0.0276
	Metabolism of cofactors and vitamins	0.7157	0.0491
	Metabolism of other amino acids	0.5042	0.0856
	Metabolism of terpenoids and polyketides	0.8628	0.0271
	Nucleotide metabolism	0.7729	0.0807
	Xenobiotics biodegradation and metabolism	0.8226	0.0309
XGBoost	Amino acid metabolism	0.4342	0.0628
with	Biosynthesis of other secondary metabolites	0.6897	0.0344
Encoding	Carbohydrate metabolism	0.6723	0.0542
	Chemical structure transformation maps	0.2580	0.0803
	Energy metabolism	0.4312	0.1211
	Glycan biosynthesis and metabolism	0.7040	0.0653
	Lipid metabolism	0.8045	0.0387
	Metabolism of cofactors and vitamins	0.5504	0.0594
	Metabolism of other amino acids	0.3589	0.0990
	Metabolism of terpenoids and polyketides	0.7717	0.0338
	Nucleotide metabolism	0.5465	0.1184
	Xenobiotics biodegradation and metabolism	0.6800	0.0428

**Table 4 metabolites-14-00266-t004:** Performance by model.

Model	Average MCC	Standard Deviation	Weighted Average MCC of Huckvale et al. [5]	Weighted Standard Deviation
MLP	0.7844	0.0129	N/A	N/A
MLP with Encoding	0.7695	0.0139	0.7240	0.1615
XGBoost	0.7637	0.0126	0.7677	0.1540
XGBoost with Encoding	0.6567	0.0152	N/A	N/A

**Table 5 metabolites-14-00266-t005:** Computational resource usage.

Model	Computational Resource	Amount
MLP	RAM (gigabytes)	2.507
	GPU RAM (gigabytes)	0.848
	Compute time (minutes)	89.706
MLP with Encoding	RAM (gigabytes)	1.764
	GPU RAM (gigabytes)	0.42
	Compute time (minutes)	129.130
XGBoost	RAM (gigabytes)	23.878
	GPU RAM (gigabytes)	11.052
	Compute time (minutes)	70.098
XGBoost with Encoding	RAM (gigabytes)	5.307
	GPU RAM (gigabytes)	5.776
	Compute time (minutes)	27.297

**Table 6 metabolites-14-00266-t006:** Top 10 most important features.

Feature/Type	Associated Pathways	Rank	Pathway Rank	Metabolite Rank	Pathway Score	Metabolite Score
C0(C0(2_(C0.10))((C0.2-1)))(C0((C0.10))) (C0((C0.10))((C0.2-1)))(C0(2_(C0.10))((C0.2-1))) Pathway	Biosynthesis of other secondary metabolites Metabolism of terpenoids and polyketides Xenobiotics biodegradation and metabolism	1	1	7042	0.206	5.225 × 10^−5^
C0(C0((C0.10)))(C0(2_(C0.10))((O0.10))) Pathway	Amino acid metabolism Biosynthesis of other secondary metabolites Carbohydrate metabolismLipid metabolism Metabolism of cofactors and vitamins Metabolism of terpenoids and polyketides Xenobiotics biodegradation and metabolism	2	2	6035	0.044	7.522 × 10^−5^
C0(C0((C0.10)))(C0((C0.10))((C2.10))((O0.20))) Pathway	Amino acid metabolism Carbohydrate metabolism Chemical structure transformation maps Lipid metabolism Metabolism of cofactors and vitamins Metabolism of terpenoids and polyketides	3	3	3905	0.0363	0.00014
O0(O0(2_(C1.10)))(C1((C0.11))((C1.10))((O0.10))) (C1((C1.10))((O0.10))((O0.16))) Pathway	Amino acid metabolism Biosynthesis of other secondary metabolites Carbohydrate metabolism Chemical structure transformation maps Energy metabolism Glycan biosynthesis and metabolism Lipid metabolism Metabolism of cofactors and vitamins Metabolism of other amino acids Nucleotide metabolism Xenobiotics biodegradation and metabolism	4	4	6265	0.02164	6.967 × 10^−5^
C0(C0((C1.10))((N0.10))) Pathway	Amino acid metabolism Biosynthesis of other secondary metabolites Carbohydrate metabolism Chemical structure transformation maps Energy metabolism Metabolism of cofactors and vitamins Metabolism of other amino acids Nucleotide metabolism	5	5	3168	0.01598	0.00017
C0(C0((C0.10))((C0.2-1))((O0.10)))(C0((C0.10))((C0.2-1))) (C0((C0.10))((C0.2-1))((O0.10)))(O0((C0.10))) Pathway	Amino acid metabolism Biosynthesis of other secondary metabolites Chemical structure transformation maps Lipid metabolism Metabolism of cofactors and vitamins Metabolism of terpenoids and polyketides Xenobiotics biodegradation and metabolism	6	6	4066	0.01134	0.00013
C0(C0((C0.10))((N0.10))((O0.20)))(C0(2_(C0.10))) (N0((C0.10))((C2.16)))(O0((C0.20))) Metabolite	Glycan biosynthesis and metabolism	7	N/A	7	N/A	0.01046
C1(C1((C2.10))((O0.10))((O0.16)))(C2(2_(C1.10)) ((N0.16)))(O0((C1.10))((C2.10)))(O0((C1.16))((P0.10))) Metabolite	Glycan biosynthesis and metabolism	8	N/A	8	N/A	0.00864
C0(C0((C0.20)))(C0(2_(C0.10))((C0.20))) Pathway	Amino acid metabolism Biosynthesis of other secondary metabolites Carbohydrate metabolism Chemical structure transformation maps Glycan biosynthesis and metabolism Lipid metabolism Metabolism of cofactors and vitamins Metabolism of terpenoids and polyketides Xenobiotics biodegradation and metabolism	9	9	1963	0.0083	0.000256
C0(C0((C0.10))((C0.21)))(2_C0((C0.10))((C0.21))) (C0((C0.10))((C0.21)))(C0((C0.10))((C0.21))((O0.10))) Pathway	Amino acid metabolism Biosynthesis of other secondary metabolites Chemical structure transformation maps Metabolism of cofactors and vitamins Xenobiotics biodegradation and metabolism	10	10	3636	0.00697	0.00015

## Data Availability

The data and code for complete reproducibility of the results in this work are available via FigShare at https://doi.org/10.6084/m9.figshare.25517695.v1.

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
