# Peer review of "Predicting the Pathway Involvement of Metabolites Based on Combined Metabolite and Pathway Features"

_metabolites, 2024, doi:10.3390/metabo14050266_

Round 1
Reviewer 1 Report
Comments and Suggestions for Authors
There is a need to develop assessment methods for predicting pathway involvement of metabolites identified in metabolomic experiments, and this study is a very welcome and prospectively valuable addition to the field. The manuscript is very well written. The main strength of this paper is that the proposed model has been reasonably well verified by aligning the model with known metabolite function. The lack of this has been a shortcoming in other recent publications in the field.
Author Response
Reviewer 1:
There is a need to develop assessment methods for predicting pathway involvement of metabolites identified in metabolomic experiments, and this study is a very welcome and prospectively valuable addition to the field. The manuscript is very well written. The main strength of this paper is that the proposed model has been reasonably well verified by aligning the model with known metabolite function. The lack of this has been a shortcoming in other recent publications in the field.
Response:
We thank the reviewer for their very positive response! We have worked hard to describe our methods and results in a clear manner. Some of the figures in the Methods section were redesigned more than once.
Reviewer 2 Report
Comments and Suggestions for Authors
The work is interesting and well conducted. However, the authors should better present and motivate their approach which seems too much elaborate and can lead to a clear overfitting of the results. Overall, I am suggesting that the paper could be improved by simplifying the methodology or better justifying its complexity, ensuring that the methods are accessible and justifiable, and demonstrating that the results are robust beyond the specific scenarios tested in the study.
Author Response
Reviewer 2:
The work is interesting and well conducted. However, the authors should better present and motivate their approach which seems too much elaborate and can lead to a clear overfitting of the results. Overall, I am suggesting that the paper could be improved by simplifying the methodology or better justifying its complexity, ensuring that the methods are accessible and justifiable, and demonstrating that the results are robust beyond the specific scenarios tested in the study.
Response:
We thank the reviewer for their positive response! While our feature and dataset engineering approach is more complex, it was required to simplify the problem to a single binary classify rather than a set of pathway-specific binary classifiers used by all prior methods. This approach also increases the number of entries in the dataset over 10-fold from the original dataset, which provides enormous benefits for model training. Also, our approach to dataset engineering has similarities to methods commonly used to craft image training and test datasets, where image transformations like rotations are used to increase the dataset size by one or more orders of magnitude over the size of the original image dataset. We have added the following to the Discussion section to frame the feature and dataset engineering within the broader machine learning field:
“The metabolite-pathway-pair approach utilizes a combination of feature and dataset engineering methods to increase the resulting dataset over 10-fold from the original gold standard dataset, improving model training and evaluation, simplifying the deployment process to a single binary classifier, and enabling the possibility to predict involvement with arbitrarily defined pathway categories.”
Reviewer 3 Report
Comments and Suggestions for Authors
Comments for abstract: The authors mentioned in lines 12-13 that 'It is common for less than half of the identified metabolites in these datasets to have known metabolic pathway involvement.' My question is, how does the developed model increase the number of known metabolic pathway involvements? Furthermore, using this model, how many identified metabolites in these datasets can be aligned with metabolic pathway involvement?"
"The authors mentioned about various prediction models but did not named anyone therefore for better understanding please mention few prediction models.
Could the authors provide numerical data indicating how much better the model developed in this study is, such as in terms of folds or percentage?
I strongly suggest that the abstract should include numerical information rather than empirical data."
Comments about introduction: "In lines 108-109, you mentioned that 'Our best model has even exceeded the combined performance of the prior benchmark models, not just in average MCC, but with a significantly lower standard deviation.' Could you please specify how many models were developed, name the best-performing model that outperformed the benchmark models, and also mention the names of the benchmark models?"
It is recommended briefly introduce the previous benchmark models in the introduction section as well.
Line 109: please first write full form of the MCC.
Comments about Figures: Please add description to each figures. please increase the font size of figure 7.
Comments about Results and Discussion: Could you please provide a comparison of your model's performance with that of previous models in terms of percentage or folds?"
Overall comments: The manuscript is well-written and contains valuable findings.
Author Response
Reviewer 3:
Issue 1:
Comments for abstract: The authors mentioned in lines 12-13 that 'It is common for less than half of the identified metabolites in these datasets to have known metabolic pathway involvement.' My question is, how does the developed model increase the number of known metabolic pathway involvements?
Response:
Technically, the model will provide predictions of metabolic pathway involvement of a certain quality, a mean MCC of 0.79. These predictions can supplement known metabolic pathway annotations for interpretative purposes like metabolic pathway enrichment analysis of metabolic annotation enrichment analysis. We have successfully used predicted metabolic annotations to derive new information from a highly untargeted lipidomics analysis that we published a few years back:
Joshua M. Mitchell, Robert M. Flight, and Hunter N.B. Moseley. "Untargeted lipidomics of non-small cell lung carcinoma demonstrates differentially abundant lipid classes in cancer vs non-cancer tissue" Metabolites 11, 740 (2021).
It was our success in this publication at predicting metabolite annotations and then using them to derive statistically significant new information that has motivated the work in this manuscript.
Issue 2:
Furthermore, using this model, how many identified metabolites in these datasets can be aligned with metabolic pathway involvement?”
Response:
The gold standard dataset we used for model training and evaluation is derived from KEGG and represents all known metabolic pathway involvement to metabolites with usable Mol file chemical representations. The mean Matthews correlation coefficient (MCC) is 0.79 for our multilayer perceptron models using the non-autoencoded dataset.
Issue 3:
"The authors mentioned about various prediction models but did not named anyone therefore for better understanding please mention few prediction models.
Response:
Respectfully, we remind the reviewer that this journal has a 200-word limit on the abstract. Also, we are emphasizing the feature and dataset engineering approach and not the specific type of machine learning model used, especially since we tested multiple machine learning methods and architectures. Given the word limit, we have prioritized addressing Issues 4 and 5 raised by the reviewer.
Issue 4:
Could the authors provide numerical data indicating how much better the model developed in this study is, such as in terms of folds or percentage?
Response:
Our best model has a MCC standard deviation of 0.013 which is an order of magnitude lower than our prior set of single pathway binary models. We have now added this to the abstract:
“We demonstrate that this metabolite-pathway features-pair approach not only outperforms the combined performance of training separate binary classifiers, but demonstrates an order of magnitude improvement in robustness: Matthews correlation coefficient of 0.784 ± 0.013 versus 0.768 ± 0.154.”
Issue 5:
I strongly suggest that the abstract should include numerical information rather than empirical data.
Response:
We have added this as follows:
“We demonstrate that this metabolite-pathway features-pair approach not only outperforms the combined performance of training separate binary classifiers, but demonstrates an order of magnitude improvement in robustness: Matthews correlation coefficient of 0.784 ± 0.013 versus 0.768 ± 0.154.”
Issue 6:
Comments about introduction: "In lines 108-109, you mentioned that 'Our best model has even exceeded the combined performance of the prior benchmark models, not just in average MCC, but with a significantly lower standard deviation.' Could you please specify how many models were developed, name the best-performing model that outperformed the benchmark models, and also mention the names of the benchmark models?"
Response:
We have done this as follows:
“We demonstrate that not only does the metabolite-pathway-pairs approach perform well compared to training a separate model per classifier, but our best model (multilayer perceptron) has even exceeded the combined performance of the prior benchmark models (Random Forest, XGBoost, and multilayer perceptron models), not just in average MCC, but with a significantly lower standard deviation.
Issue 7:
It is recommended briefly introduce the previous benchmark models in the introduction section as well.
Response:
We have done this as follows:
“We demonstrate that not only does the metabolite-pathway-pairs approach perform well compared to training a separate model per classifier, but our best model (multilayer perceptron) has even exceeded the combined performance of the prior benchmark models (Random Forest, XGBoost, and multilayer perceptron models), not just in average MCC, but with a significantly lower standard deviation.”
Issue 8:
Line 109: please first write full form of the MCC.
Response:
Thanks! We have fixed this.
Issue 9:
Comments about Figures: Please add description to each figures. please increase the font size of figure 7.
Response:
Have added the following description to the figures and increased the font size of Figure 7:
“Figure 1. Feature engineering. Starting metabolite atom coloring feature vectors are summed and normalized into pathway feature vectors. Then both metabolite and pathway feature vectors are used to train an autoencoder and generate encoded feature vectors with a reduced number of embedded features.”
“Figure 2. Dataset engineering. 5,683 metabolite and 12 pathway feature vectors are cross-joined to create 68,196 metabolite-pathway feature-pair vectors.”
“Figure 3. Overview of hyperparameter tuning, model training, and model evaluation. Optimal hyperpa-rameters are selected using 100 trials of hyperparameter tuning and then used in comprehensive model training and evaluation using 1000 CV iterations.”
“Figure 4. Performance of the MLP models by pathway category. Models using autoencoded metabolite-pathway feature-pair vectors are in orange and models using the unencoded metabolite-pathway feature-pair vectors are in blue.”
“Figure 5. Performance of the XGBoost models by pathway category. Models using autoencoded metabolite-pathway feature-pair vectors are in orange and models using the unencoded metabolite-pathway feature-pair vectors are in blue.”
“Figure 6. Overall performance by model. Models using autoencoded metabolite-pathway feature-pair vectors are in orange and models using the unencoded metabolite-pathway feature-pair vectors are in blue.”
“Figure 7. Average relative feature importance scores in descending order. Plot (a) includes all features while plot (b) only includes features with a relative feature importance of 0.0017 or greater.”
“Figure 8. Molecular structures of top 10 atom coloring features. The atoms and bonds comprising the atom coloring feature are highlighted in red.”
Issue 10:
Comments about Results and Discussion: Could you please provide a comparison of your model's performance with that of previous models in terms of percentage or folds?"
Response:
We have added the mean MCC values and their standard deviations for current best and previous best models.
“In particular, our best model demonstrates greatly improved performance robustness as evidenced by the significant reduction of the standard deviation by an order of magni-tude: mean MCC of 0.784 ± 0.013 versus 0.768 ± 0.154.”